# Reward Estimation for Variance Reduction in Deep Reinforcement Learning

**Joshua Romoff**[1,2]**, Alexandre Piché**[3]**, Peter Henderson**[1]**, Vincent Francois-Lavet**[1]**,
& Joelle Pineau**[1,2]
[1] McGill University, Montréal, Québec, Canada
[2] Facebook, Montréal, Québec, Canada
[3] MILA, Université de Montréal, Québec, Canada

## ABSTRACT

In reinforcement learning (RL), stochastic environments can make learning a policy difficult due to high degrees of variance. As such, variance reduction methods have been investigated in other works, such as advantage estimation and control-variates estimation. Here, we propose to learn a separate reward estimator to train the value function, to help reduce variance caused by a noisy reward signal. This results in theoretical reductions in variance in the tabular case, as well as empirical improvements in both the function approximation and tabular settings in environments where rewards are stochastic. To do so, we use a modified version of Advantage Actor Critic (A2C) on variations of Atari games.

## 1 INTRODUCTION

While reinforcement learning (RL) has had great successes in solving sequential decision-making tasks, high variance of some methods can make learning difficult when environments or rewards are strongly stochastic (Verbeeck et al., 2007; Henderson et al., 2018b). Several methods have been used to reduce variance, sometimes at the cost of bias. This includes generalized advantage estimation (Schulman et al., 2016), control-variates optimization (Grathwohl et al., 2017), updating the target policy via the expectation of its actions (Ciosek & Whiteson, 2018; Asadi et al., 2017), and updating the value function via the posterior mean of an estimated uncertain distribution (Henderson et al., 2017). Here, we propose a method for variance reduction by using a direct estimate of rewards $\hat{R}(s_t)$ to update the discounted value function $V_\gamma^\pi(s_t)$, rather than the sampled rewards. In the tabular case, this corresponds to using the sampled mean, while in the function approximation case this corresponds to learning a one-step value function $V_{\gamma=0}^\pi(s_t)$. We prove that this method results in theoretical variance reductions in the tabular case. We also show that it corresponds to performance gains in the tabular and function approximation settings in situations where rewards are highly stochastic.

### 1.1 BACKGROUND

We formulate our method with the fully observable Markov Decision Process (MDP). In an MDP, an agent can take an action $a_t$ based on its current state $s_t$ and receive a reward $r_t$, before transitioning to the next state of the MDP $s_{t+1}$. We focus on the discounted MDP case, where an agent tries to maximize the cumulative discounted reward $V_\gamma^\pi(s) = \left[ \sum_{t=0}^T \gamma^t r_t | s_0 = s, \pi \right]$, also known as the discounted value of a policy. It is common to learn a value estimate of the current policy via temporal difference (TD) learning, where the current estimate of the value function is used to bootstrap the next estimate according to the Bellman equation $V_\gamma^\pi(s_t) = r_t + \gamma V_\gamma^\pi(s_{t+1})$, via the loss: $\left( r_t + \gamma V_\gamma^\pi(s_{t+1}) - V_\gamma^\pi(s_t) \right)^2$. In the case of Advantage Actor Critic (A2C), the synchronous version of Asynchronous Advantage Actor Critic (A3C) (Mnih et al., 2016), a stochastic parameterized policy (actor, $\pi_\theta(a|s)$) is learned from this value estimator via the TD error. That is, the actor loss becomes: $-\log \pi(a|s) \left( r_t + \gamma V_\gamma^\pi(s_{t+1}) - V_\gamma^\pi(s_t) \right)$.

## 2  REWARD ESTIMATION

To reduce variance in the value function updates, we introduce an estimator for the reward at a given state $\hat{R}(s_t)$. In the function approximation case, learning this reward estimator becomes a simple regression problem: $\left(r_t - \hat{R}(s_t)\right)^2$. We then use this reward estimator in the TD (Sutton, 1988) update of the value function, rather than the sample reward: $\left(\hat{R}(s_t) + \gamma V_\gamma^\pi(s_{t+1}) - V_\gamma^\pi(s_t)\right)^2$. Intuitively, where rewards are highly stochastic due to environment stochasticity or due to some stochastic process in reward assignment, this estimation will reduce the variance propagated to the value function. We use the A2C algorithm for our function approximation experiments with this modified update. Diagrams of this process can be seen in Figure 1. For function approximation experiments, we build on the PyTorch implementation of Kostrikov (2018).

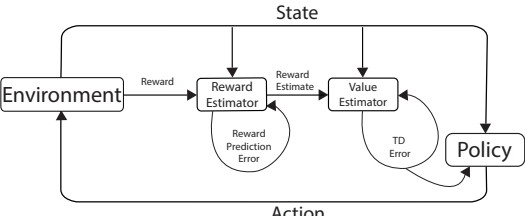

Figure 1: The actor-critic update process with the reward estimator.

**Variance Reduction in Tabular Settings:**  To determine whether our method for using a reward approximator reduces variance theoretically, we examine the tabular case. In this setting, we use the sample mean for the reward estimator: $\hat{R}_t(s) = \left[\frac{1}{N}\sum_i r_i | s_i = s\right]$. That is, given $N$ observed reward samples at a given state $s$, we determine the mean of those rewards. In this scenario, the sample mean is an unbiased estimator.

First, we determine the variance of the standard discounted Bellman equation (Bellman, 1957): $G_t^\gamma = r_t + \gamma V_\pi^\gamma(s_{t+1})$. The variance of this Bellman estimate is: $\text{Var}\left[G_t^\gamma\right] = \text{Var}\left[r_t\right] + \text{Var}\left[\gamma V_\pi^\gamma(s_{t+1})\right] + 2\text{Cov}\left[r_t, \gamma V_\pi^\gamma(s_{t+1})\right]$. We make the simplifying assumption that the most term is equal to 0 [1]. If we instead use an approximator for the reward, the Bellman equation becomes: $\hat{G}_t^\gamma = \hat{R}_t(s_t) + \gamma V_\pi^\gamma(s_{t+1})$. Similarly, the variance becomes: $\text{Var}\left[\hat{G}_t^\gamma\right] = \text{Var}\left[\hat{R}_t(s_t)\right] + \text{Var}\left[\gamma V_\pi^\gamma(s_{t+1})\right] + 2\text{Cov}\left[\hat{R}_t, \gamma V_\pi^\gamma(s_{t+1})\right]$. We note that the right most term in this case is also 0. Moreover, since we assume our approximation in the tabular case is simply the sample mean, we assume that $\text{Var}\left[\hat{R}_t(s_t)\right] = \frac{1}{N}\text{Var}\left[r_t\right]$. Thus, we have that: $\text{Var}\left[\hat{G}_t^\gamma\right] = \frac{1}{N}\text{Var}\left[r_t\right] + \text{Var}\left[\gamma V_\pi^\gamma(s_{t+1})\right]$. By re-arranging terms between the two variance estimates, we arrive at the inequality: $\frac{1}{N}\text{Var}\left[r_t\right] \leq \text{Var}\left[r_t\right]$. And thus the inequality: $\text{Var}\left[\hat{G}_t^\gamma\right] \leq \text{Var}\left[G_t^\gamma\right], \forall N \geq 1$. Therefore, by using the empirical mean of the rewards in a tabular setting, we can reduce variance. The intuitive benefit of this becomes clear in settings with highly stochastic rewards. That is, in a given state a reward may be provided with some probability $P(r)$ and otherwise is 0. In such a case, the error will propagate through longer MDP chains, whereas using the empirical mean will provide a more stable estimate, as will be demonstrated in subsequent experimental sections.

## 3  EXPERIMENTS

**Tabular Experiments:**  We first investigate the tabular case. We construct a 5 state MDP as seen for value learning (an extended 10 state MDP can be seen in Appendix A). The MDPs contain deterministic transitions from left to right in the states, and the agent follows a fixed policy moving to the right and terminates on reaching the farthest state to the right. At each state it receives a stochastic reward of 1, 2, or 5 with a fixed probability of 0.5. The value function is updated

---

[1]This holds if $V(s_{t+1})$ is conditionally independent of $r_t$ given $s_t$

via the temporal difference (TD) error for 100 episodes. We measure the robustness to variance by evaluating the root mean squared error (RMSE) of the value function across the 100 episodes. As is seen in Figure 2, when using the reward estimator, the agent is able to learn more accurate representations of the value function even at high learning rates. This indicates a parity with the aforementioned theoretical variance reduction.

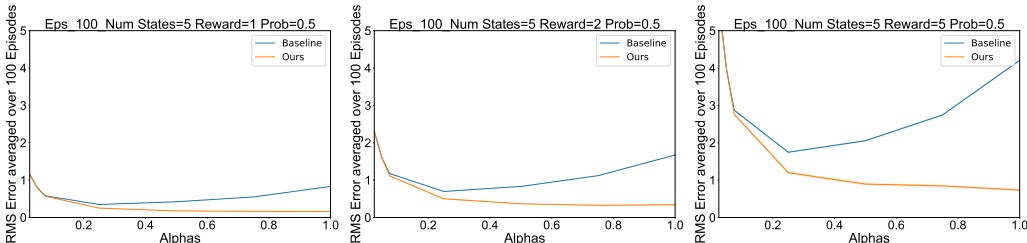

Figure 2: Tabular experiments with a 5-state MDP. In all cases, rewards are assigned with probability 0.5 and, set to 0 otherwise (rewards of $+1, +2, +5$, from left to right). The x-axis demonstrates various learning rates for the TD-update. We report the average RMSE over the first 100 episodes of learning - lower is better.

**Atari Experiments:** For evaluating our method with function approximation, we use 5 Atari games from the Arcade Learning Environment (ALE) (Bellemare et al., 2013). We experiment with adding a small amount of Gaussian noise at each time step to the reward signal (after clipping it to $[-1, 1]$). We use the exact same hyper parameters used in OpenAI's Baselines implementation (Dhariwal et al., 2017), but use an additional network (with the same architecture) as a reward predictor and use it to train our critic as described in section 2. We compare our approach to the standard A2C algorithm, as well as A2C with reward prediction as an auxiliary task, similar to Jaderberg et al. (2017). We report results, averaged over 3 random seeds, for rewards with varying levels of Gaussian noise in Table 3. For the full training curves for different amounts of noise, see Appendix B. Across all games we see that our proposed method performs relatively better once noise has been introduced.

|  | $\sigma = 0.0$ (% Baseline) | $\sigma = 0.1$ (% Baseline) | $\sigma = 0.2$ (% Baseline) | $\sigma = 0.3$ (% Baseline) | $\sigma = 0.4$ (% Baseline) |
|---|---|---|---|---|---|
| BeamRider | 122.68 | 138.76 | 392.86 | 236.78 | 170.73 |
| Breakout | 98.77 | 115.28 | 200.37 | 743.01 | 1917.97 |
| Pong | 99.78 | 121.36 | 98.50 | 1014.76 | 115.22 |
| Qbert | 63.27 | 90.16 | 173.10 | 451.56 | 546.90 |
| Seaquest | 71.93 | 91.47 | 92.07 | 157.48 | 173.08 |
| SpaceInvaders | 91.98 | 106.18 | 135.80 | 169.17 | 217.24 |
| Average | 91.40 | 110.53 | 182.12 | 462.13 | 523.52 |

Table 1: Comparison of the average episode reward over 10M steps of training between our approach to the best of both baselines (A2C and A2C with the reward prediction auxiliary task). The results are the average over 3 runs using different random seeds.

## 4 CONCLUSION

Overall, we have demonstrated that by replacing the sampled reward with reward estimator in the value function update, we can reduce variance and improve learning overall. We expect that this methodology can be particularly useful in settings with high degrees of uncertainty. To address the short-comings in some Atari games (before noise is added), future work may involve learning a distributional reward estimator as in (Bellemare et al., 2017), off-policy experience replay for the reward estimator, or learn options for reward estimators as in (Henderson et al., 2018a).

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

# A   TABULAR MDP EXPERIMENT

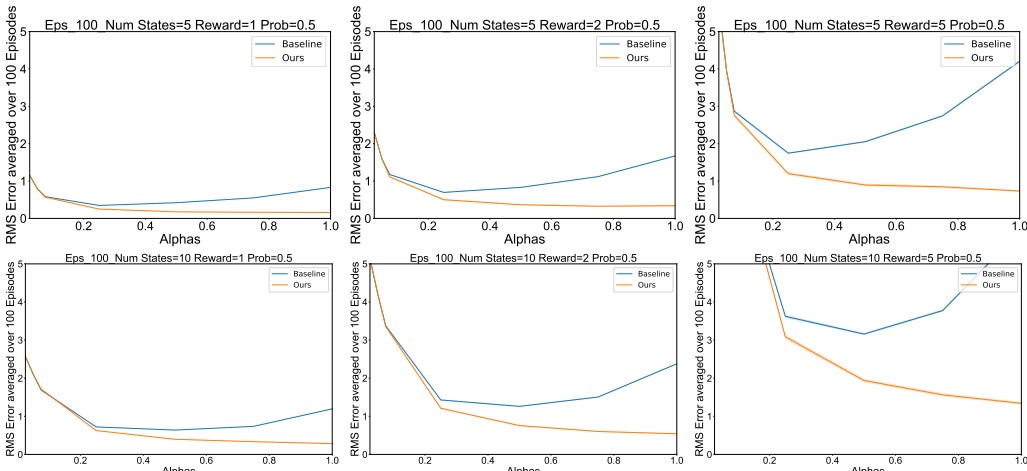

Figure 3: Tabular experiments with a 5-state MDP (top row) and a 10-state MDP (bottom row), with varying reward assignments at each states. In all cases, rewards are assigned with probability 0.5 and, set to 0 otherwise (rewards of $+1, +2, +5$, from left to right). The x-axis demonstrates various learning rates for the TD-update, as seen in similar variance analysis experiments (Van Seijen et al., 2009). As can be seen, using the sample mean in MDPs with stochastic processes greatly reduces the variance, allowing for higher learning rates to be used.

```
  0 ──────────► 1 ──────────► 2 ──────────► g
    +1 : 0, P(.5)   +1 : 0, P(.5)   +1 : 0, P(.5)
```

Figure 4: Depiction of the sample Markov Decision Process (MDP) used for the tabular case experiments.

# B  ATARI EXPERIMENTS

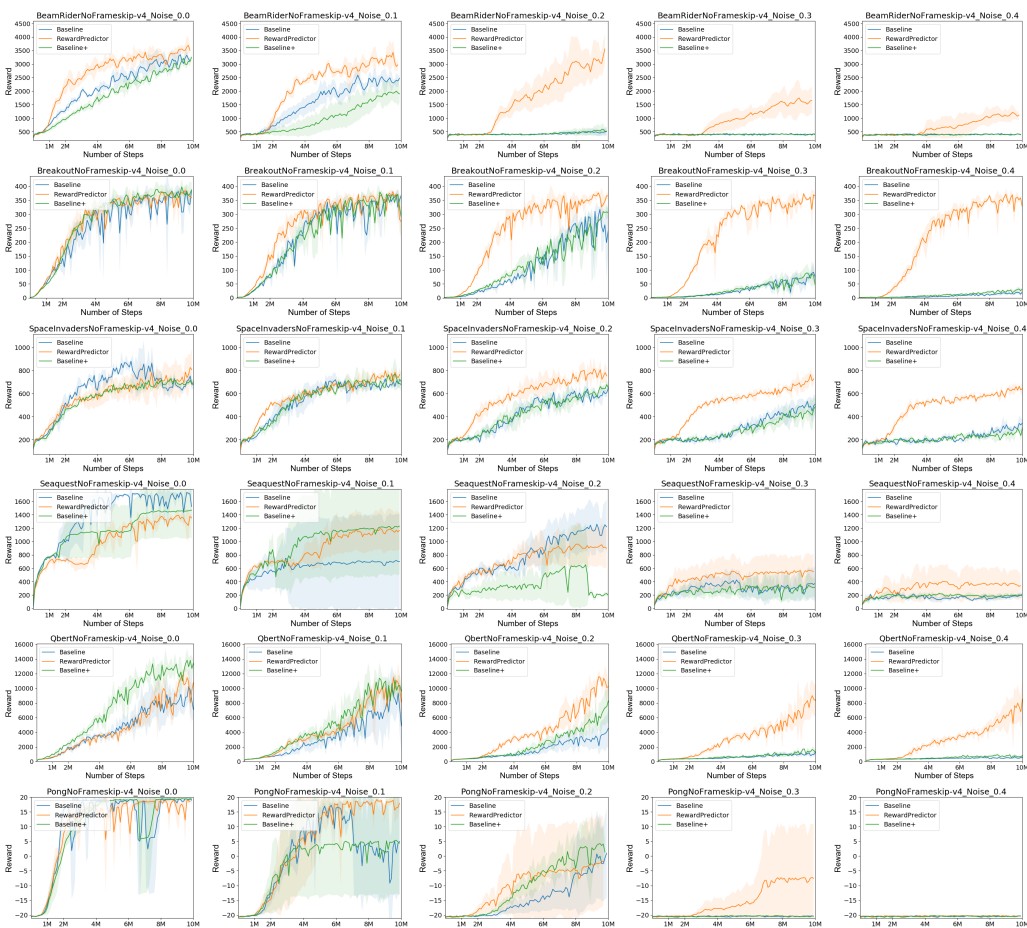

Figure 5: Full learning curves for 3 runs for five Atari Games over 10M training steps (corresponding with 40M raw frames). From top to bottom: Beam Rider, Breakout, Space Invaders, Seaquest, Qbert. We compare four different noise levels that each correspond to adding Gaussian noise centered at zero with the labelled standard-deviation. From left to right we have: 0.0, 0.1, 0.2, 0.3, 0.4. Our proposed method is labelled "Ours", while A2C and A2C with the reward prediction auxiliary task are labelled Baseline and Baseline+ respectively.

**Additional details:** As mentioned in the main text, our architecture and hyper-parameters are identical to the standard A2C parameters used in Dhariwal et al. (2017). To learn the reward-predictor, we used a completely separate network with the same overall structure as the value/policy network.

We tuned the learning rate for the reward-predictor roughly through a coarse grid-search between $[0.0001, 0.00025, 0.0005, 0.00075, 0.001]$ on a single game Pong and then used the best one $(0.0001)$ on all other games.

Additionally, we found that occasionally our algorithm diverged completely due to poor initialization of the reward-predictor. To alleviate this issue, we provided a convex combination between our estimate $\hat{r}$ and the noisy environment reward (for all environments) - which we linearly decayed over the first $25000$ network updates (out of the total $125000$ updates).

