# OpenReview forum: "Reward Estimation for Variance Reduction in Deep Reinforcement Learning"
_ICLR.cc/2018/Workshop — Accept_

### Official Review · AnonReviewer3 · 2018-03-11
**Interesting preliminary investigation**

**Rating:** 7
**Confidence:** 4

**Review:**

The paper proposes and empirically studies a simple idea, learn a reward predictor and replace the true reward in the usual advantage or Bellman equation with the predicted reward. The purpose of this is to lower variance due to randomness in the rewards. There is also a potential for an auxiliary loss benefit due to improved representations from learning to predict reward.

The authors justify the variance reduction before giving experimental results on a 10 state MDP and 5 Atari games. The reward prediction is very effective at reducing RMSE of the value function approximation in the tabular example, really surprisingly so. In Atari the results are also generally positive but less consistent than for the toy problem. Here, the authors observe that, without additional reward noise, using the predicted reward generally hurts performance. But, when rewards are noisy using the predicted reward improves performance, sometimes very significantly, in some games. There is no improvement over the baseline in QBert and Seaquest, but even in these cases the predicted-reward case seems to suffer less from the noisy rewards than the true-reward case. Perhaps there is a cost in performance due to using predicted rewards, especially prominent in these games, cause by the delayed effect of learning a value function on a learned estimate of reward?

It is a simple, self-contained, experiment and interesting result. I'm tempted to say that the approach is novel, even though reward prediction as an auxilliary loss is not, and the properties of variance being exploited similarly. There is instead maybe a novel insight being communicated here that we can exploit these methods better in RL.  I appreciated the additional experiments on Atari in the appendix, the performance of Baseline+ was the first thing I wanted to ask about after reading the main text.

It is strange that the method seems to be so minimally impacted by the reward variance, and that the two games where it performs worst are both helped (at least in the low noise case) by reward prediction as an auxiliary loss.

Pros: Simple, clear method, clearly written, doesn't hide behind complex systems so the take-away is accessible.

Cons: Results are more mixed on Atari and a better understanding of what is going on with the last two games would be useful. Could do with more work on trade-offs we get when learning on a predicted signal as opposed to the raw signal. Exploring the drawbacks to this versus the variance reduction benefits.

---

### Official Review · AnonReviewer2 · 2018-03-11
**Obvious but makes reward non stationary**

**Rating:** 5
**Confidence:** 5

**Review:**

The authors propose to learn the average reward per state in case the reward is stochastic and use this estimate instead of the actual reward in an actor critic architecture. They show it theoretically reduces variance of the value estimate.

First, I thing this idea is quite obvious. The average reward is actually what you need for the Bellman equation to hold so trying to predict that value is an obvious thing one would do if the state space is small enough.

Yet in the case of large state space, estimating the average can certainly take time and this method will make the estimated reward non stationary instead of just stochastic. The impact on stability might actually be worse than stochasticity.

The experiments are not very convincing as Gaussian noise is artificially added to Atari games natural rewards. It's a strong assumption that reward would have Gaussian distributions which is of course in favor of the method.

---

### Decision · Program_Chairs · 2018-03-20
**ICLR 2018 Workshop Acceptance Decision**

**Decision:**

Accept

**Comment:**

Congratulations, your paper was accepted to the ICLR workshop.